# The Leukemic Fly: Promises and Challenges

**DOI:** 10.3390/cells9071737

**Published:** 2020-07-21

**Authors:** Amani Al Outa, Dana Abubaker, Joelle Madi, Rihab Nasr, Margret Shirinian

**Affiliations:** 1Department of Anatomy, Cell Biology and Physiology, Faculty of Medicine, American University of Beirut, Beirut 1107 2020, Lebanon; aya11@mail.aub.edu; 2Department of Experimental Pathology, Immunology and Microbiology, Faculty of Medicine, American University of Beirut, Beirut 1107 2020, Lebanon; dma70@mail.aub.edu (D.A.); jrm06@mail.aub.edu (J.M.); 3Center for Infectious Diseases Research, American University of Beirut Medical Center, Beirut 1107 2020, Lebanon

**Keywords:** leukemia, *Drosophila melanogaster*, drug screening, fruit fly, blood cancer

## Abstract

Leukemia involves different types of blood cancers, which lead to significant mortality and morbidity. Murine models of leukemia have been instrumental in understanding the biology of the disease and identifying therapeutics. However, such models are time consuming and expensive in high throughput genetic and drug screening. *Drosophila*
*melanogaster* has emerged as an invaluable in vivo model for studying different diseases, including cancer. Fruit flies possess several hematopoietic processes and compartments that are in close resemblance to their mammalian counterparts. A number of studies succeeded in characterizing the fly’s response upon the expression of human leukemogenic proteins in hematopoietic and non-hematopoietic tissues. Moreover, some of these studies showed that these models are amenable to genetic screening. However, none were reported to be tested for drug screening. In this review, we describe the *Drosophila* hematopoietic system, briefly focusing on leukemic diseases in which fruit flies have been used. We discuss myeloid and lymphoid leukemia fruit fly models and we further highlight their roles for future therapeutic screening. In conclusion, fruit fly leukemia models constitute an interesting area which could speed up the process of integrating new therapeutics when complemented with mammalian models.

## 1. Introduction

The fruit fly *Drosophila melanogaster* has been used as a research organism to model a myriad of diseases, including cancer. Several studies used the leukemic fly to understand the transformative activity and genetic interactions of human leukemic proteins while creating a potential platform for deciphering new therapeutic targets. Hematological diseases, including leukemia or blood cancer are the result of derangements in the normal hematopoietic process. Hematopoiesis is the process whereby self-renewing multipotent hematopoietic stem cells (HSCs) differentiate into different types of blood lineages. The myeloid lineage further differentiates into different cell types, including erythrocytes, megakaryocytes (which give rise to platelets), and macrophages. The lymphoid lineage comprises B and T lymphocytes and natural killer cells. In vertebrates, hematopoiesis occurs through the primitive and definitive waves, which are spatially and temporally distinct [1]. The primitive wave is a transient wave that supports embryonic development through the production of erythrocytes by an extraembryonic yolk sac [2]. Definitive hematopoiesis is a de novo lifelong wave that gives rise to all blood cell types in mature organisms. In mammals, HSCs originate at embryonic stage in aorta/gonad/mesonephros (AGM) region of the embryo proper and homes hematopoietic organs such as the bone marrow and fetal liver [3].

Being an ectotherm that depends on external sources of heat, *Drosophila* harbors an open circulatory system with low hydrostatic pressure characterized by the presence of a simple tube-like heart (also termed dorsal vessel) and interstitial fluid known as “hemolymph.” The hemolymph is pumped from the posterior to the anterior of the fly body by the cardiac tube and it carries nutrients, metabolites, hormones, peptides, and hemocytes [4]. This review briefly introduces the hematopoietic system in *Drosophila melanogaster* and focuses on studies that used human leukemogenic proteins to demonstrate their effect on hematopoietic and non-hematopoietic fly tissues and highlights the potential role of the fly in translational leukemia research.

## *2. Drosophila melanogaster* Hematopoiesis in a Glance

### 2.1. Circulating Hemocytes and Their Response to Oncogene Expression

Major cellular immune functions in *Drosophila* are orchestrated by three types of terminally differentiated hemocytes, namely plasmatocytes, crystal cells, and lamellocytes, which are in close resemblance to the vertebrate myeloid lineage [5] and play an important role in the cell mediated innate immunity in flies [6]. Although flies are known to rely solely on their innate immunity for combatting pathogens, immune priming was described in *Drosophila* which allows for a specific immune response upon secondary infection, specifically with *Streptococcus pneumoniae* and *Beauveria bassina* [7]. Furthermore, Tasetto et al. suggested that a systemic RNAi-based adaptive antiviral response in *Drosophila* is mediated by circulating immune cells [8], indicating a resemblance to mammalian adaptive immunity with albeit distinct molecular and cellular mechanisms. Further studies in the field are required to properly characterize adaptive immunity in flies.

*Drosophila* larvae hemocytes are housed in three main compartments: the hemolymph, subepithelial patches (sessile hemocytes), and in lymph gland [9]. Accounting for 90–95% of the circulating hemocytes, plasmatocytes are considered the main representative hemocytes. They are available throughout all developmental stages with a phagocytic activity towards apoptotic debris [10] and microbes [11,12], thereby resembling mammalian macrophages/monocytes [5,10,13]. Plasmatocytes are capable of secreting antimicrobial peptides (AMPs) and hence mediate *Drosophila* humoral response [14,15]. The integration of the systemic responses and the maintenance of the organismal homeostasis necessitate the presence of a crosstalk between plasmatocytes and organs acting as barriers between the fly and its surrounding environment. Reported cross-talks include, for example, those of plasmatocytes with the gut [16] and the fat body as well as the visceral muscles [17,18,19,20]. 

The remaining ~5% of the circulating hemocytes are platelet-like cells known as crystal cells, which are non-phagocytic, harboring crystalline inclusions [21,22] that execute melanization responses, such as those required for wound healing [23] and innate immunity [24]. Crystal cells harbor prophenoloxidase, which is the essential enzyme required for melanin synthesis [25]. Both crystal cells and plasmatocytes play an important role during clotting response by secreting hemolectin, a protein that shares conserved domains present in human von Willebrand factor, coagulation factor V/VIII as well as complement factors [26,27,28]. The third type of hemocytes, known as lamellocytes, are cryptic, stress-induced cells that are rare in normal conditions and are induced in huge numbers for encapsulation of large foreign particles such as eggs of parasitic wasps during immune challenges [29]. Morphological and immuno-histochemical analysis of melanotic nodules, non-invasive “melanizations” with tumorous overgrowth in some cases, were shown to be surrounded by lamellocytes. Lamellocytes were found to mediate the encapsulation step in melanotic nodules in *Drosophila*
*hop^Tum^* and *Toll* (hematopoietic and immunity) mutants respectively [30]. Melanotic tumor or “pseudotumor” in *Drosophila*, are black melanotic spots that were reported about sixty years ago to involve hemocytes [31,32,33]. Generally, these tumours are called “melanotic masses” and, for the description of specific phenotypes, they are termed “melanotic nodules” [30]. Mutations or overexpression of Wingless (Wg), Janus Kinase (JAK)/signal transducer and activator of transcription (STAT), Toll, and Jun N-terminal kinase (JNK) are coupled to increased numbers of lamellocytes and formation of melanotic tumors [34]. For example, the constitutive activation of Toll and JAK-STAT pathways in Toll-gain-of-function/cactus-loss-of-function and hopscotch Tumorous-lethal (hop^Tum−1^) mutants respectively has been reported to result in the above-described phenotype [35,36,37]. 

Several studies demonstrated *Drosophila* hemocyte response to oncogene expression. For example, mutations of Ras genes in humans have been implicated in the pathogenesis of several leukemias [38] and interestingly in *Drosophila,* the overexpression of RasV12, an activated form of Ras, was shown to result in increased circulating hemocytes in larvae [39]. Another study expressed RasV12 in hemocytes, but using a different driver, and established two additional lines that involve RasV12 expression along with the RNAi-mediated knockdown of one of the two tumor suppressor genes lethal (2) giant larvae (l(2)gl) or scribble (scrib). Results again revealed a substantial increase in larval hemocyte numbers upon Rasv12 expression, which was more pronounced in larvae with knockdown of tumor suppressors, lethal (2) giant larvae (l(2)gl) or scribble (scrib). Interestingly, this study also revealed a deregulated immune response to oncogenic stress whereby the leukemic lines revealed activation of the Toll pathway and downregulation of the Imd pathway. Moreover, larvae coexpressing RasV12 with the RNAi construct of one of the tumor suppressors showed increased susceptibility to infections with entomopathogenic nematodes [40]. Therefore, *Drosophila* leukemia models can also be used to study both cellular and humoral arms of innate immunity. These and other studies demonstrate the response of *Drosophila* hemocytes to oncogene expression which will be further discussed in this review in Section 3. Hemocyte response includes increasing cell count, the formation of melanotic tumors and/or activating immune pathways and hence provide an interactive research platform. This is particularly important in fly models of leukemia because it provides models with a potential for genetic and/or chemical screening.

### 2.2. Drosophila Sessile Hemocytes, Stem Cells and Their Response to Oncogene Expression

Aside from the pool of circulating hemocytes, about one-third of hemocytes are described to localize to the cuticular epidermis, forming clusters known as sessile hemocytes [21]. They are comprised of differentiated hemocytes and do not home prohemocytes [41]. Sessile hemocytes derived from embryogenesis are found clustered under the larval epidermis (Figure 1) and attached to the larval imaginal discs in an organized manner [41]. In the first instar larvae, hemocytes appear as lateral patches at the lateral midline on each side of the abdominal segments. In second and third instar larvae dorsal stripes of hemocytes extend from the lateral patches. Later during the larval life, specifically in the late wandering larvae, hemocytes begin to appear on the ventral side of the larvae and the posterior dorsal vessel. Hence, sessile hemocytes are formed as the hemocytes form a sandwich between the epidermis and the muscular layer being separated from circulating hemocytes [42]. Two denser segments of resident hemocytes made up of 100–200 hemocytes, form the putative posterior hematopoietic tissue (PHT) [41]. Residency of sessile hemocytes is dependent on a trophic microenvironment provided by the peripheral nervous system (PNS). The dependency involves the colocalization and association of neuronal bodies with sessile hemocytes [42]. The exact mechanism leading to the formation of sessile hemocytes is still poorly understood [43,44,45,46]. Resident hemocytes remain in a dynamic steady state due to their ability to navigate through different clusters by active cytoskeletal rearrangements, which are disrupted upon stress resulting from immune challenges [9,42,47] or oncogene expression [40].

Previously, it was thought that adult *Drosophila melanogaster* lacks a hematopoietic organ and relies on the hematopoietic pool previously supplied by the embryo and larval hematopoiesis; until recent studies revealed active hematopoietic hubs in the dorsal abdominal region of the adult fly (Figure 1). It was shown that these hubs are dynamic and contribute to hemocyte specification and immune responses [48]. It would be intriguing to address the specific role of these sessile hubs discussed upon oncogenic stress.

An integral component of *Drosophila* hematopoietic system which can also serve as potential target for drug screening is the stem cell population. Despite the fact that HSCs are poorly understood in *Drosophila*, several stem cell populations have been identified, such as stem cells of the intestine and the ovary [49,50]. Intriguingly, upon screening for inhibitors of stem-cell-derived tumors in the fruit fly; some chemotherapeutics approved by the Food and Drug Administration (FDA) were proven to show dual effects. One effect was the reduction of the stem-cell-derived tumor growth, but another counteractive effect was inducing hyperproliferation in the wild type cells in that tumor studied. The hyperproliferation was shown to be driven by the JAK/STAT pathway, which is a highly conserved signaling pathway controlling hematopoiesis in mammals and flies. The latter results indicate that JAK/STAT may also be activated along with counteractive results in humans treated with chemotherapeutics [51]. This demonstrates the potential of *Drosophila* as a model to study stem cell population and particularly in the context of leukemogenesis. One type of stem cells that is considered to be the choice of cells to study in *Drosophila* are the stem cells found in the fly lymph gland at the posterior signaling center (PSC). A novel type of non-PSC cells was also identified in the central part of the larval lymph gland medulla extending to the cortex which exhibited stem cell characteristics [52]. These novel HSCs showed dependency on the Zinc finger protein RP-8 (Zfrp8) function. Zfrp8, homologous to the human programmed cell death protein 2 (PDCD2), functions in maintaining the identity of HSCs [52]. Other studies identified multipotent Notch expressing HSCs in the first instar larval lymph gland distinct from the long-known hemocyte progenitors. These cells are a transient cell population that expresses several molecular markers such as STAT of the conserved JAK/STAT pathway [53,54]. These HSCs serve as the founder cells for the progenitor cells of the lymph gland. Decapentaplegic (Dpp) from the PSC was involved in maintaining these novel HSCs in their niche, being identical to the vertebrate AGM HSCs [53]. Once HSCs homeostasis is lost due to a dysregulation of hematopoietic differentiation, several pathologies including leukemia may result. Since efficient tumor treatment requires the eradication of the entire stem cell population, a better understanding about CSCs and the interaction with the surrounding niche becomes important. Therefore, the fruit fly model with its primitive hematopoietic system and stem cells could serve as a powerful tool to understand the pathology of human cancers such as AML [55,56].

### 2.3. Drosophila Lymph Gland and Its Response to Oncogene Expression

In mammals, HSCs which give rise to myeloid and lymphoid lineage reside in the bone marrow, a niche which provides signals that determine blood cell renewal and differentiation. Hematopoiesis requires a dynamic communication between HSCs and their niche [57], and when this process is dysregulated, leukemias may develop [58]. *Drosophila* lymph gland and specifically the PSC offers an accessible niche that orchestrates prohemocyte differentiation and maintenance of prohemocyte in a stem cell state [59]. Although *Drosophila* hemocytes can differentiate into a limited number of cell lineages, it still presents a reductionist HSC and niche model which can be used to dissect hematopoietic processes (reviewed in [60]). Moreover, several transcription factors in the lymph gland play an important role in hemocyte development and differentiation. For example, Lozenge (Lz), a member of the Runx family of transcription factors exhibiting high homology to human AML-1/Runx1 [61], is required for development of *Drosophila* crystal cells in the lymph gland [62,63,64]. Human AML-1/Runx1 represents one of the most frequent targets subject to chromosomal translocations leading to AML [65,66]. *Drosophila* Serpent (Srp), an ortholog of the vertebrate GATA-family of transcription factors, plays a role in Lz/Glial cells missing dependent hemocyte differentiation in the lymph gland. The homology of Srp and Lz to mammalian GATA and AML1 proposes *Drosophila* as a model for dissecting mammalian hematopoiesis and leukemias [63].

The lymph gland is located approximately one-third of the larval length from the anterior end towards the dorsal side beneath the brain [67]. It develops from a set of cell cluster that arises from the cardiogenic mesoderm along with the heart-like tube, the dorsal vessel as well as nephrocyte-like pericardial cells [68,69]. A single precursor cell in the cardiogenic mesoderm gives rise to the dorsal vessel and lymph gland [58]. This resembles the mammalian hemangioblast, which can develop into both the blood and vascular cells [68,69].

It is not until the late-second to early third instar stage that the lymph gland appears as a distinct organ with the primary lobes discernable as specific structures containing variable zones (Figure 1). In addition to the architectural variation, each zone expresses its own collection of markers which is indicative of the nature of the residing hematopoietic population [69]. For example, the medullary zone (MZ), supporting hematopoietic progenitor cells, expresses E-cadherin [69], domeless [69,70] and unpaired [69,71], which are pro-hemocyte markers. The cells of the MZ quiesce, are multipotent, can give rise to all *Drosophila* blood lineages, and lack differentiation markers. Thereby, they are similar to the common myeloid progenitors (CMP) of the vertebrate hematopoietic system [72]. Progenitor cells in the MZ are maintained by PSC through signaling pathways such as the JAK/STAT and hedgehog [73,74]. The PSC, which is also the site for lamellocyte differentiation [75], expresses a set of markers such as Antennapedia [73] and Collier [75], which shares homology with mammalian Early B-cell Factor (EBF) [75,76]. Mature hemocytes reside in the cortical zone (CZ) of the lymph gland which expresses Peroxidasin [77], Lozenge [63], and Hemolectin [26].

Genetic mutations can induce tissue abnormalities that result in internal stress in *Drosophila*. If a genetic mutation disrupts normal hemocytes differentiation, signaling, and proliferation, an “auto-immune” response develops. This response can result in hemocytes attacking normal tissue [78]. Several indicators of immune activation were observed in various *Drosophila* leukemia models, where oncogenes were expressed in the hematopoietic system using specific drivers [37]. Lymph gland hypertrophy or precocious rupture, increased circulating hemocytes, lamellocyte differentiation and formation of melanotic tumors, are all characteristics of the “auto immune” response. These characteristics are additionally used as readouts for genetic screens examining possible genes involved in hematopoietic homeostasis and leukemogenesis [37,79,80,81]. Thus, the lymph gland can be regarded as an emerging model for leukemia since *Drosophila* homologues of the mammalian genes mediating leukemia have been shown to produce a “leukemia” phenotype in the lymph gland. These phenotypes include the enlargement of the lymph gland due to increased number of mature hemocytes which is similar to the increase in bone marrow cell proliferation as a result of myeloproliferative neoplasm (MPN) [82,83]. In addition, a second phenotype that occurs in the lymph gland is its early disintegration. This disintegration results from accelerated progenitor hemocytes differentiation and their release in the circulation [84]. These phenotypes were seen when pathways that mediate blood differentiation, such as adenosine signaling, Toll, and JAK/STAT, were over-activated [37,85,86]. One example is hopscotch, which encodes for the JAK kinase homologue in *Drosophila* and is implicated in human leukemias. The mutation in the hopscotch allele tumorous lethal results in the lymph gland enlargement. In addition, the hopscotch mutated lymph glands are neoplastic in nature, i.e., they can give rise to tumors upon transplantation to other adult flies [87].

The lymph gland is an important compartment to study when leukemic factors/oncogenes are introduced to fruit flies. For example, upon expressing AML associated NUP98-HOXA9 (NA9) in *Drosophila* hematopoietic system, Baril et al. observed an increase in cell proliferation in addition to hyperplasia of the lymph gland, which was correlated with dysregulated signaling in *Drosophila* homologue of the mammalian FLT-4 signaling, PVR [88]. Moreover, Giordani et al., observed premature differentiation of hemocytes and lymph gland enlargement upon feeding a Smoothened (Smo) protein inhibitor (PF-04449913) to *Drosophila* larvae. Upon administration of the same Smo inhibitor to patients, the number of leukemia-initiating stem cells in the bone marrow decreased, decreasing the chance of leukemia relapse [89]. Thus, the lymph gland can be regarded as a candidate system for researchers to screen possible genetic players involved in leukemia and to understand the molecular mechanisms of established leukemia factors.

## 3. Leukemia Models in *Drosophila melanogaster*

The hematopoietic system in *Drosophila* not only plays a critical role in responding to pathogenic invasion but also participates in clearing cancerous cells and regulating specifically conserved hematopoietic cell development and differentiation processes which are conserved between humans and *Drosophila*. For this reason, *Drosophila* has emerged as potent genetic model to study blood cell development and leukemia. In addition to the fly hematopoietic system described above which has been used in fly leukemia models, several studies used the fly non-hematopoietic tissues, most commonly, the adult fly compound eye to model human leukemogenic proteins. The eye particularly served as an in vivo tool that produces an easily identified read-out that can be seen with the naked eye. We will discuss several leukemia models below where adult fly compound eye was used to study the impact of leukemia oncogene on several processes.

### 3.1. CML Models

Studies targeting the expression of human leukemogenic proteins in *Drosophila* tissues date back to 1999, where human/fly chimeric BCR-ABL1 (p210 or p185) was expressed in *Drosophila* [90]. In chronic myeloid leukemia (CML), fusion of the breakpoint cluster region (BCR) on chromosome 22 with the Abelson murine leukemia viral oncogene homolog 1 (ABL) tyrosine kinase of chromosome 9, results in the fusion gene *BCR-ABL1*. This oncogene encodes a constitutively active tyrosine kinase (BCR-ABL1), which results in altered cellular survival, proliferation, differentiation, and adhesion properties. Most of CML patients and about one-third of Philadelphia positive (Ph+) acute lymphoblastic leukemia (ALL) patients harbor the 210-KD BCR-ABL1 [91,92]. The p-185 kD fusion protein predominates in the remaining Ph+ ALL patients and rarely in CML patients [93,94]. In the human/fly chimera BCR and the N-terminal Abl sequences were human in origin while the C-terminal Abl was *Drosophila* in origin in order to increase the likelihood of interaction with *Drosophila* proteins. The established genetic model in this study served in delineating BCR-ABL1 signal transduction whose expression in *Drosophila* CNS and eye imaginal discs resulted in CNS defects and rough eye phenotype, respectively. Both P210 and P185 could substitute for Abl in *Drosophila* Abl mutants and activated ABL pathway, but resulted in different phenotypes upon overexpression and revealed pathway activation not engaged by Abl [90,95]. In abl disabled (dab) mutant flies, which harbor a background that is more demanding for abl kinase activity, P185 but not P210 functionally substituted for Abl Kinase. While this could be attributed to the higher kinase activity of P185 or the lethality of flies induced by longer BCR sequences in P210, it is clear how signaling in these fly tissues revealed the transformative potential of these 2 oncogenes and their important functional differences. Flies expressing P210 and P185 also showed higher phosphorylation of the cytoskeletal regulator enabled (ena) as compared to flies expressing *Drosophila* Abl (dAbl), which is reminiscent of the engagement of BCR-ABL1 in disrupting cytoskeletal proteins in mammalian systems [96]. Transgenic flies harboring chimeric BCR-ABL1 were not further explored for their potential in contributing to the leukemia field only until recently when the full human BCR-ABL1 is used. Our lab, following Bernardoni et al., used transgenic flies harboring the full human BCR-ABL1 fusion protein to further explore the use of fruit flies both as a genetic or drug screening model for CML. We as well as Bernardoni et al. showed that full human BCR-ABL1 (P210) expression in *Drosophila* eyes, results in a rough eye phenotype indicating the transformative potential of the fusion gene. Bernardoni et al. [97] showed that the induced eye phenotype is kinase-dependent and is most likely due to BCR-ABL1 interaction with endogenous dAbl signaling pathway such as leading to increased phosphorylation of the dAbl substrate Ena. Not only was BCR-ABL1 shown to modulate endogenous interactors in the fruit fly eye epithelium but also a *Drosophila* homolog of STAT5, a gene that is known to be implicated in BCR-ABL1-induced leukemogenesis [98], was shown to modulate the phenotype. Testing the sensitivity of the model in a more leukemia representative tissue, the lymph gland, Bernardoni et al. showed that BCR-ABL1 expression results in increased proliferation of hemocytes. The hematopoietic phenotype was again affected by dAbl signaling pathway. All of this further demonstrates the strength of this model in studying genetic interactions in CML. While this study demonstrated how a *Drosophila* BCR-ABL1 model could be exploited for studying genetic interactions in CML; we went further to test whether the system is sensitive to pharmacologic inhibition using the eye phenotype as an efficient phenotypic read-out [99]. We were able to identify a particular defect in BCR-ABL1 induced eye phenotype that showed high sensitivity to pharmacologic inhibition by tyrosine kinase inhibitors (TKIs), which are currently employed in treating CML patients. The model did not only demonstrate sensitivity to drug reversal of abnormal phenotypes but also the real difference in effectiveness and potency among the used TKIs. Potent TKIs such as dasatinib and ponatinib revealed a significantly higher tendency and efficiency than the less potent imatinib and nilotinib in rescuing the abnormal phenotype and restoring ommatidial development. Such a model serves as a highly efficient primary screening assay to filter out potential hits from a large number of compounds in drug libraries followed by validation in mammalian hematopoietic models. Another important finding in our study is linked to the expression of BCR-ABL1^p210/T315I^ mutation which revealed a more severe eye defect than BCR-ABL1^p210^ hinting to a unique signaling signature of the T315I mutation. Future experiments targeting this possibility can unveil important therapeutic targets to be considered in the case of T315I mutation. Collectively, the use of the fruit fly as a CML model is still in its infancy but holds much potential to the leukemia field. Further validation of the model using fly tissues/cells that recap the properties of the hematopoietic system, such as lymph gland and hemocytes is required to derive more relevant conclusions in the CML field.

### 3.2. Insights into Other Leukemia Models in Drosophila Melanogaster

Another type of leukemia that is the most studied in the fruit fly model is AML. AML is the most common leukemia after chronic lymphocytic leukemia and the most important cause of leukemia-associated deaths in the US [100]. Despite the progress in characterizing the pathophysiological events behind myeloid malignancies, only few drugs are approved by the FDA for AML treatment [101]. AML involves the proliferation of undifferentiated hematopoietic precursor cells and is often associated with non-random chromosomal translocations, therefore, impairing crucial hematopoietic regulators [102]. The translocation (t8;21) is the most common chromosomal abnormality that yields the *AML1-ETO* fusion gene encoding AML1-ETO chimeric protein [103,104,105,106,107]. AML1, also known as the *Runx1* gene, is a member of the runt domain (RD) family of transcription factors and whose DNA binding domain is fused to ETO in Runx1-ETO fusion protein. The mechanism by which fusion of AML1 to ETO alters AML1 function has been investigated in the fruit fly in non-hematopoietic and hematopoietic tissues. Lozenge *lz,* Runx1 homologue in the fly, is studied for its role in fruit fly eye [108] and crystal cell [63] development. The fly eye served as an in vivo system to demonstrate how AML1-ETO acts as a constitutive transcriptional repressor of *lz* target genes [109]. Using the fly hematopoietic system, two studies worked at the same time to investigate the effect of AML1-ETO expression. A study by Osman et al. [79] used *lz-gal4* to direct the expression of AML1-ETO to *Lz*^+^ blood cell lineage and found that AML1-ETO interfered with the differentiation of RUNX^+^/crystal cells. This is reminiscent of its effect in mammalian systems [110] whereby cells were maintained in a progenitor state, and their proliferation ability was increased. AML1-ETO expression under the effect of *lz-gal4* also resulted in pupal lethality, which was exploited as a sensitive phenotype for identifying suppressors in a genetic screen. The RNAi based screen identified the protease calpainB and AAA+ ATPase RUVBL1/Pontin [111] as required for AML1-ETO induced phenotype in crystal cells. Both calpainB mutation and down-regulation were proven to interfere with AML1-ETO induced phenotypes, thereby incorporating this protease as a requirement for AML1-ETO function in *Drosophila* RUNX+ blood cells. Validation of these findings in a mammalian model has shown that treatment of AML cells harboring the (t8;21) translocation and constitutively expressing AML1-ETO with calpain inhibitors resulted in decreased cell viability and clonogenicity. Therefore, the regulation of AML1-ETO seems to be conserved from *Drosophila* to humans and this proposes calpain inhibitors as a potential therapy in AML.

The second study [80] used *hemolectin* (*hml^Δ−^gal4)* to direct the expression of AML1-ETO to *Drosophila* circulating hemocytes and documented hyperproliferation of these cells along with the expansion of hemocyte precursors and the formation of melanotic tumors. The phenotype was shown to be dependent on elevated ROS levels and to involve the DNA-binding capacity of the oncogene and its interaction with cofactor CBFß as well as transcriptional repressors. Moreover, suppressors and enhancers of AML1-ETO-induced melanotic tumor formation and hyperproliferation of hemocytes were identified through screening 231 genomic deficiencies on *Drosophila* autosomes and 1500 autosomal insertion mutations and represent interesting candidates to be validated in mammalian systems.

Collectively these two concurrent studies revealed that an AML1-ETO *Drosophila* model recaps much of the leukemogenic properties and behavior of the fusion gene in mammalian systems. AML1-ETO in mouse models was shown to increase self-renewal of progenitor cells as means to suppress myeloid differentiation [112], and this was demonstrated in the fruit fly model as altered differentiation of crystal cells [79].

Some fruit fly AML models involved the myeloid leukemia factor (*MLF*), which is a family of conserved genes coding for small proteins that act in nucleo-cytoplasmic shuttling. In patients with myelodysplastic syndrome (MDS) and AML, human MLF (hMLF1) was characterized as a target of the t(3;5) (q25.1;q34) translocation producing the fusion protein between the entire hMLF1 and the N-terminal domain of nucleophosmin (NPM1) [113]. Interestingly, *Drosophila mlf* was demonstrated to stabilize Lz as means to regulate lz+ cells and to maintain the lymph gland homeostasis keeping the cells in a progenitor state. Moreover, human MLF1 was shown to be able to substitute for *Drosophila mlf* through rescuing the crystal cell defects in *mlf* mutants. Both human MLF1 and *Drosophila mlf* were shown to be necessary for the expression of RUNX1-ETO in a stable manner in human and *Drosophila* leukemia cells, respectively [114]. The importance of this finding lies in the fact that no RUNX factor has been linked to progenitor state homeostasis. Therefore, MLF should be interacting with new mediators which, if further analyzed provides deeper knowledge of controlling the progenitor fate of cells.

Two different leukemogenic fusions that involve *MLL* (Mixed Lineage Leukemia) gene and occur in 5–10% of AML or ALL cases were studied in *Drosophila* as well. AF4 and AF9 are transcription factors that constitute the most common fusion partners to MLL, forming MLL-AF4 and MLL-AF9, respectively [115]. The former occurs exclusively in ALL cases, and the latter is associated with AML [116]. Transgenic flies expressing MLL were shown to be completely viable, whereas those with the expression of human MLL-AF9 and human MLL-AF4 fusion proteins showed larval/pupal lethality. Although the chimeric MLL proteins showed the same phenotype, they appeared to act distinctly. Both showed altered cellular division but those expressing AF9 revealed an expedited rate of cell proliferation, whereas AF4 revealed a delayed one. Moreover, MLL-AF9 but not MLL-AF4 flies exhibited significant chromatin aberrations, and both displayed different binding patterns on polytene chromosomes, which indicate that they target distinct genes. The chimeric proteins MLL-AF4 and MLL-AF9 harbor a C-terminus that replaces PHD fingers and SET domains of MLL and appears to play a role in shaping the activity of MLL by differentially targeting specific genes [117]. Hereby, *Drosophila* can serve in this context to further elucidate the distinct pathways followed by the two fusion proteins contributing to leukemogenesis.

Human NUP98-HOXA9 (NA9), which is a translocation that fuses the amino-terminal of NUP-98 to the carboxyl-terminal of HOXA9, was also studied in *Drosophila melanogaster*. Homeobox (HOX) genes are known to affect cellular decisions of self-renewal and differentiation [118], and along with their cofactors such as MEIS and PBX are considered as frequent targets of epigenetic and genetic AML modifications [119]. Both NUP-98 and HOXA9 have been shown in murine models to lead to AML, whose onset was vastly hastened by MEIS1 co-expression [120,121]. By exploring the effect of human NA9 on *Drosophila* hematopoietic system using *hml*-gal4, researchers identified an increase in hemocytes count along with lymph gland hyperplasia, which resembles the effect of NA9 in a murine model [88]. The phenotype required NUP98 moiety as well as the PIM and HD domains of HOXA9; most importantly, HTH, the fly homolog of MEIS, was shown to cooperate with NA9 reminiscent of its role in a murine model. Interestingly, expressing NA9 in the cortical zone and circulating hemocytes resulted in PSC expansion, which highlights the effect of NA9 on the lymph gland niche and shows how *Drosophila* can act as a model to study the interaction of leukemogenic proteins with their microenvironment.

Epigenetic derangements, involving improper histone methylation, constitute important targets to consider when investigating potential AML treatments. Histone mutations were shown to take place early throughout the leukemogenic process and to contribute to the main leukemic clone in AML patient samples [56,122]. *Drosophila* was used in this context as a model organism harboring a simple hematopoietic system to confirm the effects of histone mutations on hematopoiesis. An increase in the number of circulating hemocytes was detected in larvae overexpressing H3K27M mutated histones as compared to wild type larvae, thus specifically indicating the effect of histone mutations on cellular immune response in *Drosophila* and further elucidating the crucial role of H3K27M mutations [56].

Adult T-cell leukemia/lymphoma (ATL), which occurs secondary to HTLV-1 (human T-cell lymphotropic virus type 1) infection, was also modeled in *Drosophila* [123]. HTLV-1 encodes the transactivator Tax-1, which is crucial for cellular transformation [124]. This study used both fly hematopoietic and non-hematopoietic tissues to study Tax-1 induced phenotype and interacting partners in vivo. The fly eye was used to demonstrate the transformative phenotype of Tax-1, which revealed disruption of the normal ommatidial arrangement resulting in a rough eye phenotype. Using peroxidasin-gal4 Tax-1, transgenic flies exhibited a significantly higher number of circulating hemocytes. The eye phenotype was exploited for a genetic screen to identify Tax-1 genetic interactors. It is noteworthy to mention that the expression of Tax-2, which is encoded by non-oncogenic HTLV-2, did not cause any phenotypes in the eye or hemocytes, further ensuring the efficiency and sensitivity of this model [125]. One of the crucial targets of Tax-1 is known to be NF-κB pathway, which drives the transformation of HTLV-1 infected T cells [126]. The RNAi-mediated knockdown of Relish, the *Drosophila* NF-κB family member, and Kenny, the *Drosophila* orthologue of IKKγ/NEMO, resulted in a significant reduction of the Tax-1 induced rough eye phenotype. This indicates that Tax-1 induced transformation in *Drosophila* is dependent on NF-κB pathway activation, and this provides a valuable transgenic model to study Tax-1 mutants further as well as demonstrate how HTLV-1 tax transforms cells in vivo.

### 3.3. Potential Avenues for Using the Leukemic Fly Model for Drug Discovery

Murine models are considered powerful tools for elucidating the pathogenesis of leukemia as well as for deciphering potential therapeutics and therapeutic targets. However, several disadvantages also accompany these models [127], particularly when it comes to high throughput drug screening. In this context, *Drosophila melanogaster* provides an inexpensive, genetically, and molecularly tractable non-mammalian model that is amenable to high-throughput screening methods [128]. Chemical screening has been successfully carried out using fruit fly models for a myriad of human disorders such as those inflicting the kidney, metabolism and central nervous system as well as for multiple endocrine neoplasia type 2A and 2B (MEN2) and lung cancer (reviewed in [55]). Recently the platform for drug screening has been pushed to involve personalized fly models that recap the specific genetic aberrations of cancer patients [129]. Although leukemia has been modeled elegantly in flies, none of the reported models was validated for drug screening except for our study on CML. The fruit fly models described in this review, and summarized in Table 1 provide valuable approaches for drug discovery if adapted to such methods. Our research group validated the fly eye as an efficient primary screening assay for drug screening in CML and we are currently working on validating the results in the fly hematopoietic system. Several AML models are available (Table 1). AML is considered as fatal for about 80% of the patients [100] and fly models should be further characterized and tracked for their suitability as drug screening platforms, which could help identify potential therapeutic targets for this virulent type of leukemia. The described studies in Table 1 highlight the importance of such models in identifying crucial genetic aberrations that take place in AML by performing RNAi-based genetic screens, which further encourage the use of the models for identification of potential treatments. We notice that most of the work in the AML field is focused on deciphering the genetic interactions, and this is highly explicable in light of the lack of understanding of the mechanisms that inaugurate the cancerous state of AML cells. However, genetic and drug screening should go hand in hand for more efficient results in this domain.

When it comes to the utilization of fly leukemia models in drug screening, drug delivery to the site of action as well as, the nature of the disease shapes important restraints. When using the GAL4/UAS system to target the expression of the oncogenes to a specific fly tissue, researchers should search for the optimal phenotype that would allow for drug reversal effects. This might necessitate a change in temperature of culturing the flies since the GAL4/UAS system is temperature-sensitive. Another important notion is the tissue where the oncogene is expressed, a screen that includes different target tissues might be desirable for picking the one that shows sensitivity to drug action. Although the fruit fly bestows at our hands several tissues that range from eyes to CNS, hemocytes and the hematopoietic niche represented by lymph gland remain the tissues of choice that can be exploited for modeling leukemia. These compartments provide a closer representation of the actual derangements occurring in human leukemias. Therefore, results extracted from non-hematopoietic tissues should be consequently validated in the fly hematopoietic system before embarking on further analyses while bearing in mind that the fly hemocytes have a closer resemblance to the mammalian myeloid lineage.

The fruit fly with its different dynamic hematopoietic compartments, changing throughout the fly development, provides a valuable in vivo model that simplifies the complexity of the mammalian bone marrow for studying associated leukemias. We summarize in Figure 1 some of the phenotypes to study upon modeling leukemic genes in the fly. The larval stages with its main hematopoietic compartment, the lymph gland, along with circulating and resident hemocytes, which are all encased in a transparent cuticle constitute an attractive in vivo model. Most of the times, the phenotypes can be easily visualized even by the naked eye, thus contributing to the efficiency of any attempted screen. Circulating hemocytes can, for instance, be counted to indicate any proliferative phenotype, stained with specific markers for identification of any differentiation imbalance, and even tested for activation of specific pathways in question. Sessile or resident hemocytes show a neat pattern that can be examined for any disturbance upon oncogene expression. Although the larval lymph gland disintegrates upon metamorphosis, the adult fly was shown to still harbor a bone marrow like niche consisting of hemocytes homed to hematopoietic hubs and hence can be exploited during leukemia studies. Adult flies, on the other hand, provide a highly elegant arrangement of ommatidia in the eye structure, which have been used as an efficient and easy primary read-out for genetic and chemical screening.

Resistant mutations in different types of leukemia can hinder the success of standard treatments. The fly, with its easily manipulated genes, and the availability of genetic tools and resources in the fly community, can be efficiently used in this context.

Collectively, the fruit fly leukemia models still hold much potential for contributing to our understanding of unresolved aspects of leukemia pathogenesis as well as to speed up the identification of potential treatments. Data generated from high throughput screening in flies can be easily validated in more complex mammalian models.

## 4. Conclusion and Perspective

While *Drosophila* leukemia models that we and others presented recapitulate some of the evolutionary conserved pathways, mediators, and molecules found in human and fly hematopoietic systems, there still exists a number of unresolved and challenging questions. For example, although *Drosophila* lymph gland can be used to study hematopoietic cell lineage, proliferation, differentiation, and the signals required to maintain a progenitor cell in its undifferentiated form, it still lacks a well characterized hematopoietic stem cell that fully resembles the mammalian one. Furthermore, since both innate and adaptive immune sensing mechanisms are required in anti-tumor response in most if not all hematological malignancies, the lack of a homologous adaptive immune arm makes the fruit fly a less comprehensive model. As discussed earlier in this review, it is only recently that hematopoietic hubs were discovered in adult flies and it will be interesting to understand how these hubs respond to oncogene since most leukemia studies have only looked at larval responses. Moreover, with the advancement in genome editing, it is now possible to create *Drosophila* avatars where multiple genes can be modified, thus better resembling the genetic complexity of leukemia patients [130]. This will not only be of importance to unravel mechanisms of oncogene interaction that drives leukemogenesis, but will also present as an efficient tool to screen for better therapies. *Drosophila* remains an exceptionally simple and powerful tool to identify genetic interactors of oncogenic protein and to understand the signaling mechanism behind oncogene regulation at both cellular and molecular levels. However, more efforts are required in translating these findings into higher organisms and humans, which must be made in parallel to the advancement of our knowledge of fly hematopoiesis.

## Figures and Tables

**Figure 1 cells-09-01737-f001:**
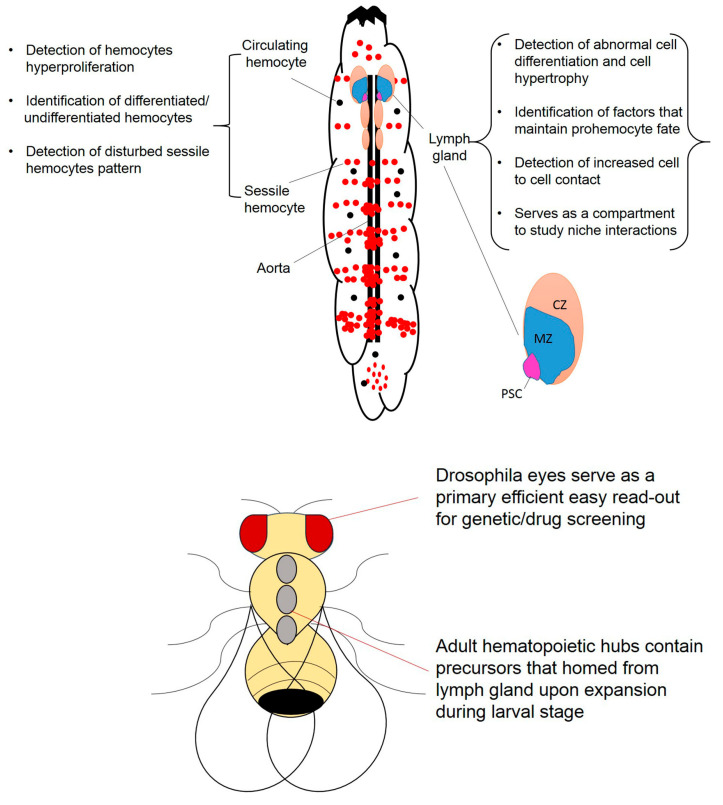
Summary of the hematopoietic compartments in fruit flies that can be used for leukemia studies. *Drosophila* larva serves as an efficient in vivo model with its easily accessible hematopoietic compartments and transparent cuticle that allows clear identification of oncogene-induced phenotypes. The adult fly serves in providing a primary efficient read-out for genetic and chemical screening. The larval lymph gland and hematopoietic hubs in the larvae and adult respectively shape a simple mammalian bone marrow that can be exploited in leukemia research. MZ: medullary zone (blue); CZ: cortical zone (orange); PSC: posterior signaling center (purple).

**Table 1 cells-09-01737-t001:** Summary of human leukemia models in *Drosophila melanogaster.* This table summarizes the studies that used *Drosophila melanogaster* for modeling human leukemogenic proteins in hematopoietic and non-hematopoietic tissues and their major findings.

Type of Leukemia	Transgene	Phenotype	Site of Oncogene Expression	Reference
CML /Ph^+^ ALL	Human/fly BCR-ABL1^P210^ /BCR-ABL1^P185^	- CNS and eye defects and increase in the phosphorylation of the dAbl substrate “Ena”	CNS and eye imaginal discs	[90]
		- embryonic lethality and disruption of morphogenesis (disruption of head involution, segment grooves and dorsal closure)	Various embryonic sites	[95]
	Human BCR-ABL1^P210^/ BCR-ABL1^T315I^	- Altered differentiation in *Drosophila* eyes and interference with dAbl signaling - Increase in circulating hemocytes	Eye imaginal discs and hemocytes	[97]
		- T315I resulted in a more severe rough eye phenotype - The model was validated for drug screening by feeding flies TKIs	Eye imaginal discs	[99]
ALL/AML	MLL, MLL-AF9, and MLL-AF4	- MLL-AF9, and MLL-AF4 cause larval/pupal lethality upon expression in blood lineage and during early and late development - The fusions showed differing effects on proliferation and chromosome condensation in larval brain	Ubiquitously, all imaginal discs and in hematopoietic system	[117]
AML	Human AML1-ETO	- AML1-ETO acts as a transcriptional repressor of Lozenge target genes in *Drosophila* eyes	Eye imaginal discs	[109]
		- Expression in Lz+ blood cells inhibited the differentiation of crystal cells, and induced an increase in circulating Lz+ progenitors - Identification of calpain B as required for AML1-ETO activity in *Drosophila* hemocytes	Hemocytes	[79]
		- In vivo RNAi in *Drosophila* expressing human AML1-ETO identifies Pontin/RUVBL1 as a gene responsible for AML1-ETO-induced lethality and blood cell proliferation	Hemocytes	[111]
		- Expression in majority of circulating hemocytes using (*hml*-Gal4) increased hemocytes count and along with expansion of hemocytes progenitors	Hemocytes	[80]
	Human MLF1	- *Drosophila mlf* appeared to play a role in RUNX1-ETO stabilization - Human MLF1 expressed under the control of *lz*-Gal4 reversed *mlf*-associated crystal cell defects	Crystal cell lineage	[114]
	Human NUP98-HOXA9 (NA9)	- Expression of NA9 in *Drosophila* cortical zone of lymph gland and circulating hemocytes results in increased cellular proliferation and enlargement of posterior signaling center	Lymph gland and hemocytes	[88]
ATL	HTLV-1Tax transactivator (Tax-1)	- Eye defects and increased circulating hemocytes - Knockdown of Relish of the IMD pathway reversed the rough eye phenotype through an RNAi-based screen	Eye imaginal discs and hemocytes	[123]

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
