# Peer review of "The Leukemic Fly: Promises and Challenges"

_cells, 2020, doi:10.3390/cells9071737_

Round 1

Reviewer 1 Report

This is an interesting review article addressing fruit fly as an animal model to study the oncology. It is well written and comprehensible. I think the research fellows should know more about this opportunity in labs.

Author Response

We thank reviewer 1 for his/her positive feedback on our review.

Reviewer 2 Report

The recent progress in understanding leukemic pathophysiology has increased the effort to identify drugs able to target leukemogenic proteins. In this paper the authors propone fruit flies as models for drug screening in alternative to the traditional more expensive and time consuming mouse models.

In the first part of the paper they focus on fly hemato-lymphopoietic components describing the different terminally differentiated hemocytes, their “physiologic” function, their response to stress or oncogenic trigger and the genes involved in neoplastic transformation, underlying the frequent similarity to human leukemogenic pathways.

Moreover, they report that a novel type of HSC, identified in the central part of the larval lymph gland medulla, express genes involved in stem cell maintenance and could be used to study stem cell homeostasis in their niche and as a consequence the potential targets to eradicate leukemia stem cells.

Finally, they suggest that response to oncogenic trigger could be efficiently evaluated in the different population of lymph glands, in which stress-induced tissue abnormalities produce  a leukemia phenotype, as a result of activation of the same signally pathways involved in human leukemogenesis, making these cells an excellent candidate to test new target therapies.

In the second part of the paper the authors summarize the current knowledge on leukemogenesis arising from the use fly non-hematopoietic tissues.

In chronic myeloid leukemia Drosophila models contributed to better understand the different leukemia phenotype resulting from the different abl activity, but they also gave information on the real difference and potency among TK inhibitors.

In different acute leukemia subtypes, fruit-fly models provided information on perturbed intracellular pathways and permitted to identify gene cofactors, potential target for new drugs.

The paper is very interesting, especially in the first part, well “organized” and clear. Figures and tables are adequate, references are updated.

Author Response

We thank reviewer 2 for his/her positive feedback on our review manuscript. 

Reviewer 3 Report

In summary:

The manuscript is rather confusing and very difficult to understand.

Several of the statements are not correct and some data are not cited correctly.

Furthermore, the hematopoietic system of fruit flies is not just a simplified blood cell system of the vertebrate system.

Since the authors claims to concentrate on CML and AML in this review, they should give a better overview and comparison of the myeloid lineages in flies and vertebrates. In addition, they should point out the profound differences between flies and vertebrates concerning the immune system.

Specific comments:

Page 2, line 51, whole paragraph

The whole introduction is not very structured. The authors jump between different topics, for example mentioning lymphoid lineage cells, which are important for the vertebrate system but not for invertrebrate, but this isn’t addressed any further. Also it isn’t pointed out, that leukemia is described by massive proliferation of hematopoietic progenitor cells and what this means when drosophila cells are used to study leukemia.

I have problem to understand several sentences in this paragraph, here are some examples:

Page 2, line 52:

What is meant with oncogenic trigger?

Page 2, line 75 What does that mean?: Lamellocytes are also involved in melanotic tumor formation?

Page 2, line 86 What does that mean?: Several studies demonstrated Drosophila hemocyte response to an oncogenic trigger.

Page 3, line 95: What does that mean?: Interestingly, this study also revealed a deregulated immune response to oncogenic stress whereby the leukemic lines revealed activation of Toll pathway and downregulation of Imd pathway. ?

Page 3, line 103

Again, what does oncogenic trigger mean?

Page 3, line 103

Reference 45 refers to EBV DNA injection (not the whole virus!), not to oncogene or microorganisms

Page 4 , Figure 1

Abbreviation missing for the lymph gland

Page 4, line 138:

What is meant with oncogene-induced stress? Why do the authors now talk about the cancer stem cells?

Page 5-6, Paragraphs 2.3 and 2.4

The paragraphs are a random description of the structural anatomy, it is unclear to me, what role the cells play for a leukemic model in the fruit fly

Page 10, line 407

Why now T-cell leukemia?

Page 10, line 433

What is a personalized fly model?

This unstructured and unprecise phrasing is found in the whole manuscript.

Author Response

We would like to thank reviewers for their constructive comments and advise which improved our review. In the following we will address the reviewers’ comments and point out the changes incorporated to the manuscript and figure in line with their suggestions. Please note that all changes in the manuscript are indicated in red.

Reviewer 3:

The manuscript is rather confusing and very difficult to understand. Several of the statements are not correct and some data are not cited correctly. Furthermore, the hematopoietic system of fruit flies is not just a simplified blood cell system of the vertebrate system. Since the authors claims to concentrate on CML and AML in this review, they should give a better overview and comparison of the myeloid lineages in flies and vertebrates. In addition, they should point out the profound differences between flies and vertebrates concerning the immune system.

To improve the structure of the review we have clarified all the points raised by reviewer 3, modified the citations accordingly and added further explanations to statements which were unclear.  We agree with the reviewer that the “hematopoietic system of fruit flies is not just a simplified blood cell system of the vertebrate system” Therefore; we have deleted “and provide a simplified version of the latter” from line 17 in the abstract.

Specific comments:

  1. Page 2, line 51, whole paragraph

The whole introduction is not very structured. The authors jump between different topics, for example mentioning lymphoid lineage cells, which are important for the vertebrate system but not for invertebrate, but this isn’t addressed any further. Also it isn’t pointed out, that leukemia is described by massive proliferation of hematopoietic progenitor cells and what this means when drosophila cells are used to study leukemia. I have problem to understand several sentences in this paragraph, here are some examples:

The whole introduction is not very structured. The authors jump between different topics, for example mentioning lymphoid lineage cells, which are important for the vertebrate system but not for invertebrate, but this isn’t addressed any further.

We thank the reviewer for pointing this out. We have added a paragraph indicating what is described in the literature (Pham, L.N et al; 2007 and Tassetto, M et al; 2017) regarding adaptive immunity in fruit flies and that this field still requires further investigations. We have added the following explanation (page: 2, Lines: 55-61): Although flies are known to rely solely on their innate immunity in combatting pathogens, immune priming was described in Drosophila suggesting specific immune response upon secondary infection specifically with Streptococcus pneumoniae and Beauveria bassina [7].  Furthermore; Tasetto et al. suggested the presence of  systemic RNAi-Based adaptive antiviral response in Drosophila which is mediated by circulating immune cells [8]; indicating a role for  adaptive immunity in flies  albeit distinct  from the mammalian system both at the molecular and cellular levels.  

 leukemia is described by massive proliferation of hematopoietic progenitor cells and what this means when drosophila cells are used to study leukemia.

Several studies in Drosophila melanogaster expressing human leukemia oncogenes described distinct phenotypes in hemocytes upon oncogene expression such as: an increase in circulating hemocytes upon conditional expression of BCR-ABL1 in the medullary zone of lymph gland which contains prohemocytes (Bernardoni et al., 2019); impaired crystal cell differentiation upon expression of AML1-ETO in LZ+/RUNX+ cell lineage (Osman et al., 2009). Furthermore; an increase in the number of circulating hemocytes was reported upon the expression of AML1-ETO fusion gene in Drosophila hemocytes (Sinenko et al., 2010) as well as altered hemocytes differentiation leading to excessive production of hematopoietic precursors. These phenotypes are recapitulated in mouse models in which AML1-ETO suppresses myeloid differentiation by increasing self-renewal of progenitors. We have addressed this aspect of oncogene expression and hemocyte proliferation and differentiation on page 8, lines: 309-311 (Bernardoni et al., 2019), page: 8 lines 348-350 (Osman et al., 2009) and page: 9 lines: 362-364 (Sinenko et a., 2010l) and  

Page 2, line 52: What is meant with oncogenic trigger?

What we meant here was in fact the stress associated with tumorigenesis upon oncogene expression. For the sake of clarity and in line with the reviewer concerns, we have replaced the term oncogenic trigger in the whole manuscript with more precise terms/explanation which we indicate below:

Page: 2, line: 51: We have deleted “to an oncogenic trigger” and added “to oncogene expression”

Page: 3, line: 96: We deleted “to an oncogenic trigger” and added “to oncogene expression”

Page: 3, line: 110: We rephrased the sentence to: “These and other studies demonstrate the response of Drosophila hemocytes to oncogene expression”

Page: 3, line: 116: We rephrased the title to: “Drosophila sessile hemocytes and stem cells and their response to oncogene expression”

Page: 5, line: 178:  We have deleted “to an oncogenic trigger” and added “to oncogene expression”

Page 2, line 75 What does that mean?  Lamellocytes are also involved in melanotic tumor formation?

We meant to indicate that lamellocytes play a role in the encapsulation step of melanotic nodule formation. This was shown in a study by (Minakhina et al., 2006) where recruitment of lamellocytes to melanotic nodules was reported in some fly mutants. This is now clarified in the text page: 2 lines: 83-87.

Page 2, line 86 What does that mean? : Several studies demonstrated Drosophila hemocyte response to an oncogenic trigger.

 As previously mentioned we meant the response to stress associated with tumorigenesis upon “oncogene expression” which is now added instead of “oncogenic trigger “in the text. Page: 3, Line: 96.

What does that mean?: Interestingly, this study also revealed a deregulated immune response to oncogenic stress whereby the leukemic lines revealed activation of Toll pathway and downregulation of Imd pathway. ?

We thank the reviewer for raising this point and as per his/her suggestion we have added a concluding sentence to indicate the importance of Drosophila leukemia models in identifying which innate immune pathways whether cellular or humoral are activated upon various oncogene expression.   

This was discussed in a study conducted by Arefin et al.The Immune Phenotype of Three Drosophila Leukemia Models.” where the authors expressed RasV12 in hemocytes alone or together with RNAi-mediated knockdown of one of the two tumor suppressor genes lethal (2) giant larvae (l(2)gl) or scribble (scrib).  In addition to cellular immune response mediated by Drosophila hemocytes upon expression of RasV12 or suppression of tumor suppressors, humoral immune pathways which are mediated by Toll and Imd, similar to TLR and TNFR signaling in mammals respectively, were also affected. Toll pathway was activated and Imd pathway downregulated. Moreover; larvae coexpressing RasV12 with RNAi knock out of one of the tumor suppressors showed increased susceptibility to infections with entomopathogenic nematodes. We now added to the revised manuscript a concluding sentence (page: 3, lines: 108-110) to clarify this idea: therefore highlighting how Drosophila leukemia models can be also used to study both cellular and humoral arms of innate immunity.”

Page 3, line 103: Again, what does oncogenic trigger mean?

We also refer here to “oncogenic expression” which is now replaced in the revised text: page: 3, line: 116.

Page 3, line 103: Reference 45 refers to EBV DNA injection (not the whole virus!), not to oncogene or microorganisms

As the reviewer indicated, we were referring to the EBV DNA and not the virus but for the sake of clarity we modified the text accordingly by deleting “microorganism” and incorporating the EBV reference (ref # 45) to references indicating stress resulting from immune challenges page 3 line 133.

Page 4, Figure 1: Abbreviation missing for the lymph gland

We have added the abbreviation of lymph gland to the figure legend page: 4, lines: 147-148” MZ: medullary zone (blue); CZ: cortical zone (orange); PSC: posterior signaling center (purple).”

Page 4, line 138: What is meant with oncogene-induced stress? Why do the authors now talk about the cancer stem cells?

For the sake of clarity and to avoid confusion we removed the sentence regarding cancer stem cells along with the reference related to it page: 4, reference number 49. We have also modified the description of stem cells and removed “oncogene-induced stress” and its related reference number 48.

Page 5-6, Paragraphs 2.3 and 2.4: The paragraphs are a random description of the structural anatomy, it is unclear to me, what role the cells play for a leukemic model in the fruit fly

We thank the reviewer for this comment and in line with his/her concerns we have added an introductory paragraph explaining the role of the lymph gland in leukemia fly models:

In mammals, HSCs which give rise to myeloid and lymphoid lineage reside in the bone marrow, a niche which provides signals that determine blood cell renewal and differentiation. Hematopoiesis requires a dynamic communication between HSCs and their niche (59) and when this process is dysregulated leukemias may develop (60). Drosophila lymph gland and specifically PSC offers an accessible niche that orchestrates prohemocyte differentiation and maintenance of prohemocyte in a stem cell state (61). Although Drosophila hemocytes can differentiate into a limited number of cell lineages, it still bestows in our hands a reductionist HSC and niche model which can be used to dissect hematopoietic processes (reviewed in 62). Moreover, several transcription factors in the lymph gland play an important role in hemocyte development and differentiation. This is now added to page: 5, lines: 179-187

Page 10, line 407: Why now T-cell leukemia?

We have modified the abstract as per the reviewer’s comment and added lymphoid leukemia (lines 21-22) since we are also discussing lymphoid leukemia, although more briefly, and mentioning it in our table page 13.

Page 10, line 433: What is a personalized fly model?: This unstructured and unprecise phrasing is found in the whole manuscript.

This term has been used in page: 10, line: 453 and is clarified in the phrase following it in line: 454 “fly models that recap the specific genetic aberrations of cancer patients”. This term is used in a study by Bengi et al., entitled: “A personalized platform identifies trametinib plus zoledronate for a patient with KRAS-mutant metastatic colorectal cancer”. Whereby in this study the authors established a Drosophila model that reflected the complexity of a colorectal cancer patient’s genome. The model was produced by alterations of Drosophila orthologs of nine genes that were identified in the patient’s genome in the fruit fly. The personalized construct that was generated for the patient expressed a GAL4-inducible (i) UAS-ras85DG12V transgene and (ii) synthetic eight-hairpin cluster targeting the Drosophila orthologs of eight tumor suppressor genes. Screening for drug candidates that can reverse lethality in flies identified trametinib, a Ras pathway inhibitor, and zoledronate (a bisphosphonate) as drug candidates then the patient was treated with this combination.

Round 2

Reviewer 3 Report

The authors have improved the structure of the manuscript.

Understandably, as experts for drosophila the authors focus on the hematopoietic system of the fly. Therefore, even after revision, this review is of interest to researchers in the specific field. To researchers from the field of oncology who wish to extend their knowledge to use alternative models for leukomogenesis it is still not very comprehensive.

Author Response

We thank the reviewer for his/her feedback and we agree with him/her that this review might comprehensively describe almost all aspects of fly hematopoiesis as well as all blood cancers that were studied in Drosophila which will be of interest to the fly community.  In order to extend the knowledge to the broader oncology field in this current version we made sure that each section and subsection always reflects on the relevance when it comes to comparisons in humans and leukemia. Changes are made in red color.

  • We have deleted from pages: 5, 6 and 7, lines: 202-211, 214-218, 230-239, 261-270 from the lymph gland part which contained a very detailed anatomical description of the lymph gland and instead added relevant mechanisms and TFs that are functionally conserved between fruit flies and humans in hematopoiesis (page: 5 , line: 187-194, 221-229)
  • We have added a small transition paragraph before moving on to the leukemia models studied in flies (page: 7, lines: 283-288)
  • We have added a “Conclusion and Perspective” section to summarize some of the shortcomings of this system in studying leukemia and emphasize on the attributes of Drosophila that still makes it a powerful model to understand leukemogenesis better (page: 14, lines: 535-556)
